# THINK YOU HAVE SOLVED COMMONSENSE REASONING? TRY HELLASWAGULTRA

## ABSTRACT

With the evolution of large language models (LLMs), widely used commonsense reasoning and natural language understanding benchmarks have become saturated. At the same time, the number of languages supported by LLMs has been growing rapidly, while existing benchmarks cover only a limited set of languages, leaving many unsupported. Moreover, some multilingual benchmarks rely on translating English benchmarks, which introduces evaluation bias. To address these issues, we propose HellaSwagUltra, a commonsense reasoning and natural language understanding benchmark covering 60+ languages. It includes a large amount of local cultural knowledge for each language. We design an automated data construction pipeline, making it easy to continuously expand. Unlike existing work that explicitly tests reasoning skills, HellaSwagUltra embeds two commonsense or local knowledge facts implicitly in the context of each question. Each answer choice reveals subtle clues indicating whether the knowledge is violated. Models must sensitively detect these differences between options to select the most plausible continuation. In addition, we recruited experts for each language to fully review and correct all test items, and we continue to update them. Experiments show that even the strongest proprietary models (e.g., Gemini-2.5-Pro) achieve only 62.5% accuracy, while GPT-4o and leading open-source models remain near 40–50%. Our results highlight that multilingual commonsense reasoning remains a major open challenge, and we release both dataset and pipeline to support future research. Our data is anonymously open at `https://anonymous.4open.science/r/xjQkRbtWnhsu-2F86`.

## 1 INTRODUCTION

The trajectory of large language model (LLM) development shows a decisive move toward multilinguality. Contemporary academic and commercial models increasingly advertise competence in dozens or even hundreds of languages, moving past the era where English dominated generative model deployment. For instance, Gemma3 (Team et al., 2025) claims coverage of more than 140 languages, while Qwen3 (Yang et al., 2025) reports support across 119 languages and dialects. Similarly, closed-source systems such as ChatGPT (OpenAI et al., 2024), Claude [1], and Gemini (Team et al., 2024a) promote their strong multilingual performance, though the exact scope of their linguistic coverage is not publicly detailed.

However, multilingual evaluation has not kept pace with the rapid progress of large language models. In particular, multilingual commonsense reasoning evaluation remains underexplored. A major challenge lies in the scarcity of data: for languages other than English, both the quality and quantity of available data lag far behind. Although many efforts have expanded multilingual evaluation sets by translating existing English benchmarks (Lai et al., 2023; Huang et al., 2025; Singh et al., 2025), this approach suffers from translation quality issues and cultural bias. To address these limitations, several natively multilingual test suites have been proposed, such as CMMLU (Li et al., 2024), IN-CLUDE (Romanou et al., 2024) and MultiLoKo (Hupkes & Bogoychev, 2025). Nevertheless, these datasets are primarily drawn from native wiki documents or exam questions in each language, they tend to focus on factual knowledge assessment and overlook the most crucial aspect of multilingual evaluation — natural language understanding and commonsense reasoning.

---

[1] `https://www.anthropic.com/news/claude-4`

A second challenge is that commonsense reasoning benchmarks are harder to construct than knowledge-based test sets. Unlike knowledge benchmarks, there are no large pools of ready-made questions, so they often require scenario creation, plausible distractors, and nuanced human annotations. Moreover, commonsense questions rarely have a single definitive answer, making careful human judgment essential. Widely used commonsense reasoning benchmarks such as Hellaswag (Zellers et al., 2019), StoryCloze (Mostafazadeh et al., 2016), CommonsenseQA (Talmor et al., 2019) have become saturated, with strong large language models already achieving near-perfect scores ($> 90\%$) on them. Continuing to expand them through human annotation is costly and makes it difficult to achieve higher levels of challenge. Figure 1 shows the example from HellaSwag that is facing the saturation issue. In multilingual settings, the challenge is amplified because commonsense knowledge is contextual and culturally dependent, requiring additional effort to ensure that questions remain valid and fair across languages. Although several multilingual benchmarks have been introduced (Li et al., 2025; Sakai et al., 2024b; Ismayilzada et al., 2023), they cover only a limited set of languages and pay insufficient attention to local culture and social context.

A third challenge arises in difficulty design, especially in multilingual settings. In terms of difficulty design, existing work often focuses on the task format (Li et al., 2025; Xiong et al., 2025; Ismayilzada et al., 2023) — such as causal reasoning, multi-hop reasoning, abduction (reasoning from effect to cause), and ordering tasks. To some extent, these measure a model's ability to follow instructions. As a result, base models or smaller models tend to collapse in performance under such complex formats, making it difficult to accurately reflect their fundamental language understanding ability. Consequently, current commonsense reasoning benchmarks often show low scores for small models but near-saturation for strong models. This limits their usefulness for guiding LLM pre-training or fine-tuning, as performance tends to exhibit sudden jumps rather than gradual improvement.

To address these challenges, we introduce HellaSwagUltra, a benchmark for multilingual commonsense reasoning that spans over 60 languages and is grounded in each language's native culture, social context, and commonsense knowledge. This broad and culturally diverse coverage directly tackles the lack of suitable multilingual benchmarks. To overcome the inherent difficulty of constructing commonsense datasets, we adopt the natural language inference format of HellaSwag, which aligns closely with the training objective of causal LLMs and enables stable evaluation even for smaller models. We further design a fully automated pipeline: starting from culturally relevant Wikipedia pages, we prompt LLMs to extract commonsense, generate structured consequences and subtle violations, and then roll these into narrative contexts. This automation makes it possible to scale data construction while maintaining consistency. Finally, to address the difficulty of designing fair and informative evaluation in multilingual settings, we embed two commonsense intents implicitly in each story and construct distractors that subtly contradict them. This ensures that the benchmark requires careful reasoning about plausibility, while avoiding the saturation of strong models and the collapse of smaller ones. Table 1 summarizes the comparison with existing work, and the full list of supported languages is provided in Appendix A.

Table 1: Comparison of commonsense benchmarks. **NLI** refers to task types where the subsequent content is inferred from the preceding text; these tasks generally preserve the fluency of natural language corpora. **QA** refers to simple question-and-answer formats. **Constructed** refers to tasks composed of more complex, human-defined setups, typically including an instruction. Task difficulty is determined by GPT-4 accuracy: **\*** $\geq 80\%$, **\*\*** $\geq 50\%$, **\*\*\*** $< 50\%$

| Benchmark | Supported Languages | Total Samples | Local Commonsense | Task Format | Difficulty |
|---|---|---|---|---|---|
| HellaSwag | En | 10k | ✗ | NLI | * |
| StoryCloze | En | 1.8k | ✗ | NLI | * |
| CommonsenseQA | En | 1.1k | ✗ | QA | * |
| mCSQA | 8 | 11k | ✗ | QA | * |
| CRoW | 5 | 16k | ✗ | Constructed | * |
| Com^2 | En | 3.7k | ✗ | Constructed | ** |
| HellaSwag-Pro | 2 | 12k (Zh) | ✓ | Constructed | * |
| HellaSwagUltra (ours) | 61 | 60k+ | ✓ | NLI | *** |

To summarize, our contributions are as follows:

**HellaSwag**

**Context:** Then, the man writes over the snow covering the window of a car, and a woman wearing winter clothes smiles. then …

**Continuations:**

**A.** the man adds wax to the windshield and cuts it.

**B.** a person board a ski lift, while two men supporting the head of the person wearing winter clothes snow as the we girls sled.

**C.** the man puts on a christmas coat, knitted with netting.

**D.** the man continues removing the snow on his car. ✅

Overly simplistic assessment intent. ☹

Excessively pronounced option disparities. ☹

**Com^2**

**Question:** What interventions can help prevent negative outcomes in the scenarios described?

Emma, a aspiring actress, joins a local community theater troupe ... Emma's casual drinking escalates into unhealthy habits, harming her relationships and well-being.

**Options:**

**A.** Encourage regular group discussions about sobriety and setting limits on alcohol consumption. ✅

**B.** Organize recreational activities that do not include alcohol, such as game nights or hiking trips. ✅

**C.** Increase the number of rehearsals to ensure every actor knows their lines perfectly.

**D.** Focus on promotional materials that emphasize the glamorous lifestyle of acting to attract more talent.

Reliance on instruction following. ☹

Explicit commonsense expression. ☹

**HellaSwagUltra**

**Context:** In the dim light of the kitchen, Leo pulled a ceramic plate of cold leftovers from the refrigerator. He placed a metal fork on the food, slid the plate into the microwave, and set the timer for one minute before immediately cancelling it. From the doorway, Chloe watched him wordlessly take a single egg from the carton on the counter. He placed the whole egg inside the now-empty microwave, shut the door, and set it to cook for two minutes as the pile of dirty dishes sat by the sink.

**Continuations:**

**A.** A loud pop erupted from inside the microwave, and Leo stood motionless, staring at the humming appliance. Chloe flinched at the sound, her hands tightening into fists at her sides as she took a sharp breath. ✅

**B.** Bright flashes arced from the fork for a moment before he hit cancel. When the timer beeped, he removed the egg and peeled the shell away over the sink, revealing a firm, solid white.

**C.** A loud pop sounded from the microwave; Leo opened the door, placed the plate with the fork into the splattered interior, and set the timer. When the cycle finished, he opened the door to a cloud of steam rising from the food.

**D.** When the microwave beeped, Leo peeled the shell away from the egg's firm, solid white. He then put the plate with the fork on it back into the appliance, and a minute later opened the door to a cloud of steam rising from the food.

**Embedded Commonsense:**

- Metal in a microwave will generate sparks.

- Heating a whole egg in a microwave will cause it to explode.

Advanced commonsense evaluation. ☺    Multi-commonsense integration. ☺    Implicit commonsense embedding. ☺

**Context:** 林悦紧锁眉头，指尖在手机屏幕上划过，对身旁几次欲言又止的陈默置若罔闻。他脚边的礼品袋随着车厢的轻微震动而摇晃。陈默拿出手机，点开那两张电子车票，再次确认了"预定到站时间：上午10点30分"的字样。他收起手机，目光无意识地落在车厢前方的电子显示屏上，上面滚动的实时速度数字刚刚跳过300。 *Lin Yue's brows were tightly furrowed, her fingertips sliding across the screen of her phone, ignoring Chen Mo's several attempts to speak. The gift bag at his feet swayed slightly with the faint vibration of the carriage. Chen Mo took out his phone, opened his two e-tickets, and once again confirmed the words "Scheduled arrival time: 10:30 a.m." He put his phone away, his gaze drifting unconsciously to the electronic display at the front of the carriage, where the scrolling real-time speed had just passed 300.*

**Continuations:**

**A.** 车厢广播开始播报到站信息，电子显示屏上的时间变为上午10点30分。列车平稳地滑入站台，停稳的瞬间，他身旁小桌板上半瓶水的水面才泛起一丝涟漪。 *The carriage broadcast began announcing the arrival information, and the time on the electronic display changed to 10:30 a.m. The train glided smoothly into the station, and only at the moment it came to a complete stop did the surface of the half-full bottle of water on the tray table beside him ripple slightly.* ✅

**B.** 列车开始减速，林悦手机旁那杯满水的纸杯，水面依旧平稳。列车滑进站台时，陈默的视线扫过对面站台的电子屏，上面显示一趟预定10点40分出发的列车状态为"已发车"。 *The train began to slow down, but the surface of the full paper cup of water next to Lin Yue's phone remained perfectly still. As the train glided into the platform, Chen Mo's gaze swept across the electronic display on the opposite platform, which showed that a train scheduled to depart at 10:40 had already "Departed."*

**C.** 车厢里响起一阵规律的"咣当、咣当"声。前方显示屏上的时间变为上午10点30分，列车广播开始播报，车身同时滑入站台。 *A rhythmic "clack-clack" sound echoed through the carriage. The time on the display screen at the front changed to 10:30 a.m., the train's announcement began to play, and the train glided into the platform at the same time.*

**D.** 车厢里周期性地响起"咣当、咣当"的声响。列车减速滑入站台，他们走出车门时，对面轨道空无一物，一块显示着"10:40"字样的电子牌刚刚熄灭。 *A periodic "clack-clack" sound echoed through the carriage. As the train slowed and glided into the platform, they stepped out of the door to find the opposite track completely empty, and an electronic sign displaying "10:40" had just gone dark.*

**Embedded Commonsense:**

- China's high-speed railway tracks are continuously welded, so there is almost no jolting inside the carriage even at high speeds.

- In China, high-speed trains are generally very punctual.

Story scenarios aligned with the linguistic and cultural background. ☺    Local commonsense knowledge. ☺

Figure 1: Existing commonsense benchmarks are reaching saturation and cover only a limited set of languages, with insufficient focus on language-specific local commonsense. HellaSwagUltra spans 60+ languages, incorporates a wide range of local, culturally grounded commonsense scenarios, embeds commonsense knowledge implicitly in the context, and offers highly challenging distractor options.

- We introduce HellaSwagUltra, the first large-scale multilingual commonsense reasoning benchmark covering over 60 languages and 62k instances, explicitly grounded in local cultural knowledge.

- We design a fully automated construction pipeline that scales across languages while addressing the intrinsic difficulty of generating realistic scenarios, subtle distractors, and consistent annotations.

- We propose a difficulty scheme that embeds multiple implicit commonsense facts in each context, ensuring stable evaluation across both small and strong models and mitigating the saturation observed in prior benchmarks.

- We release HellaSwagUltra-Gold, a human-verified subset for high-stakes evaluation, and provide extensive experimental results showing that even state-of-the-art LLMs remain far below human performance.

## 2 RELATED WORK

**Multilingual Benchmarks**   Existing multilingual benchmarks can be roughly divided into two categories. The first category relies on translating and extending English benchmarks. BenchMax (Huang et al., 2025) expands a diverse set of benchmark tasks from English into 17 languages, covering multiple language families, but its coverage of commonsense reasoning remains limited. MuBench (Han et al., 2025) focuses on widely used English benchmarks for pretraining evaluation — including some commonsense reasoning and natural language understanding benchmarks like HellaSwag (Zellers et al., 2019), StoryCloze (Mostafazadeh et al., 2016), SNLI (Bowman et al., 2015), and MultiNLI (Williams et al., 2018) — extending them to 61 languages and offering flexible evaluation formats. However, these commonsense reasoning benchmarks are already close to saturation, suffer from significant data contamination issues, and carry risks of cultural bias. Beyond English-based extensions, a second line of work collects native-language corpora. INCLUDE (Romanou et al., 2024) gathers exam questions from 44 languages, emphasizing assessment of local, language-specific knowledge. MultiLoKo (Hupkes & Bogoychev, 2025) extracts fact-based question–answer pairs from Wikipedia articles in multiple languages, posing a high level of difficulty. Nevertheless, these benchmarks focus primarily on factual knowledge evaluation rather than broader commonsense reasoning.

**Commonsense Reasoning Benchmarks**   In addition to classic benchmarks such as HellaSwag (Zellers et al., 2019), StoryCloze (Mostafazadeh et al., 2016), and CommonsenseQA (Talmor et al., 2019), several recent efforts have sought to increase task complexity in order to mitigate performance saturation on commonsense reasoning evaluations. Com^2 raises the difficulty by constructing complex causal graphs and defining multiple task types. HellaSwag-Pro (Li et al., 2025) similarly decomposes causal relations to create challenging tasks such as backward reasoning and ordering. However, these approaches often disrupt the natural coherence of the original text, which can lead to unstable evaluation results. Moreover, complex task formats mainly test instruction-following rather than genuine understanding of language and commonsense. On the multilingual side, there has also been work supporting commonsense reasoning across languages. HellaSwag-Pro (Li et al., 2025) introduced a new Chinese dataset constructed through self-bootstrapping, while mCSQA (Sakai et al., 2024a) extracts aligned concepts across languages from ConceptNet. They covers only a small number of languages and lacks high-quality, in-depth local commonsense knowledge.

## 3 HELLASWAGULTRA

HellaSwagUltra adopts the simplest task format — selecting the most plausible continuation given the provided context. This task format preserves the semantic coherence of the text and aligns with the training objective of causal LLMs, allowing it to accurately and reliably reflect a model's natural language understanding capability. The construction process of HellaSwagUltra consists of several key stages: **Knowledge Extraction**, **Structured Commonsense Generation**, **Test Rollout**. Figure 2 depicts the data collection pipeline.

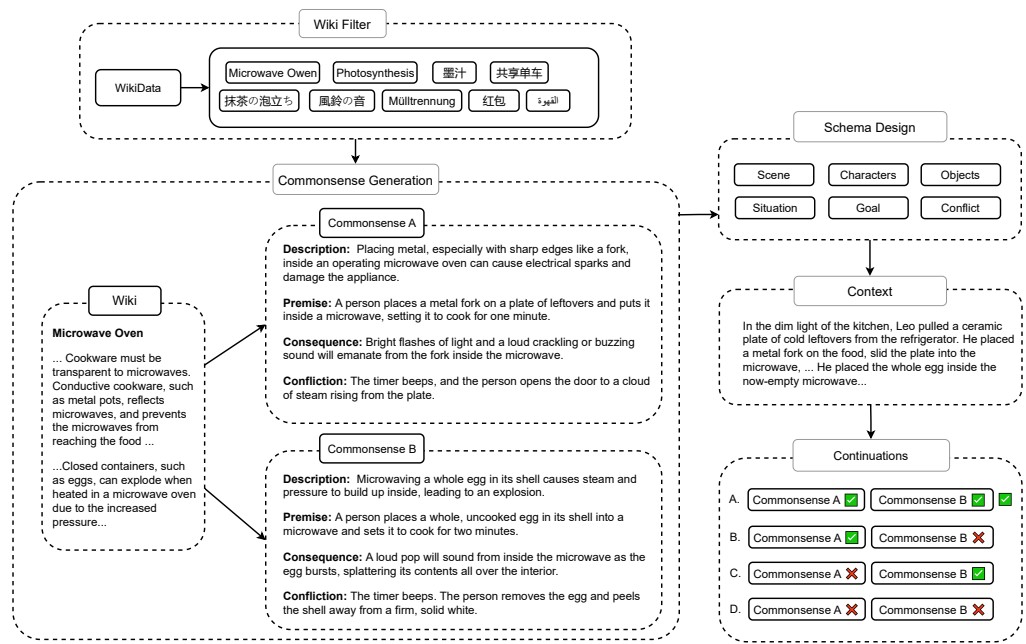

Figure 2: The automatic data construction process consists of three main stages: Knowledge Extraction, Structured Commonsense Generation, and Test Rollout. The Test Rollout stage itself is composed of Schema Design and Context and Continuation Generation.

## 3.1 KNOWLEDGE EXTRACTION

To obtain local commonsense knowledge for each language, we filter entities from Wikidata [2] that are relevant to the local cultural and social background of each language. The filtering process consists of two steps:

**Heuristic Filtering** We first filter entities based on their QIDs. A predefined type pool is used to exclude broad categories of entities that are not useful for commonsense extraction, such as persons, organizations, geographic locations, and dates. All entities that have an instance of or subclass of relationship with these categories are removed.

**LLM-Based Filtering** For the remaining candidate QIDs, we feed the corresponding wiki titles and pages to an LLM [3], asking it to determine whether the article is niche, whether it contains potential commonsense knowledge, and whether it carries a risk of bias. If the article passes these checks, we collect its pages in all available languages and prompt the LLM once again to identify which pages represent entities and articles tied to the local background of that specific language.

For each language, we select approximately 1,000 Wiki entities and articles to guide and control the LLM in generating targeted commonsense knowledge.

## 3.2 STRUCTURED COMMONSENSE GENERATION

This is the core stage of our entire pipeline, where we must extract deep, culturally grounded commonsense knowledge for each language to embed into the stories generated later. We design carefully crafted prompts to accomplish this, incorporating multiple iterations and validity checks. The Wiki pages collected in the previous step are provided to the LLM, which is tasked with generating structured commonsense that includes a **Description**, **Premise**, **Consequence**, and **Conflict**.

---

[2]https://www.wikidata.org/wiki/Wikidata:Main_Page

[3]We used Gemini-2.5-Pro throughout the entire data construction process.

**Description**   The LLM is instructed to produce a concise one-sentence description of each commonsense instance. This field serves two purposes: facilitating later human review and enabling automatic deduplication. We compute semantic embeddings of the descriptions using an embedding model and calculate the inner product between each newly generated commonsense description and all previously collected ones. Commonsense instances with excessively high similarity scores are discarded.

**Premise**   This field instructs the LLM to create a premise that establishes the condition under which the generated commonsense applies, based on the commonsense description. The generation follows two principles: **Sufficiency and Necessity**: The occurrence of the premise should deterministically lead to a consequence, and the consequence's occurrence must imply that the premise has taken place. **No Outcome Leakage**: The premise must not contain or reveal the consequence itself.

**Consequence**   Based on the commonsense description and the specified premise, LLM generates a correct consequence.

**Conflict**   The LLM is required to generate a detail that subtly implies a violation of the commonsense, without making it too obvious, in order to increase the difficulty of the question. The design principles are as follows:**No Direct Negation**: The conflict must not be a direct negation of the people, objects, or events mentioned in the premise or consequence. **Objective Description**: The conflict should describe the scene or event objectively, avoiding ambiguous statements or speculation, and must not include characters' subjective thoughts or feelings. **Intrinsic Plausibility**: the conflict itself must be reasonable and cannot involve surreal or impossible events.

As shown in Figure 2, the generation of a single question requires two distinct commonsense instances. After Commonsense A is successfully produced, it is added as a reference to the demonstrations. The LLM is then tasked with generating Commonsense B. In addition to following all the previously defined principles, Commonsense B must satisfy an additional independence principle: **Logical Independence** — that is, whether B is violated should not affect the judgment of whether A is violated. This prevents the LLM from producing two similar commonsense statements, which would otherwise reduce the challenge and discriminative power of the answer options.

To further ensure the quality of the generated commonsense, we employ an LLM-based validator to check compliance with each requirement. The principles defined above are compiled into a ten-item checklist. For every newly generated commonsense instance, the validator is prompted to answer each question: it must respond "Yes" if the requirement is satisfied, and "No" with an explanation if it is not. The explanations are logged and incorporated into the next generation prompt, instructing the LLM to revise its output. This process is repeated iteratively until the commonsense passes all items on the checklist.

### 3.3   TEST ROLLOUT

**Story Schema Design**   After obtaining the commonsense pairs, we do not use them directly to generate the story context. This is because giving the commonsense pairs to the LLM as-is would cause the model to overemphasize them in the story, making the commonsense too obvious. This not only reduces the diversity and naturalness of the stories but also lowers the overall difficulty of the questions. We first provide the LLM with only the premises of the two commonsense instances and ask it to design a structured story schema based on the information they contain. The design follows these principles: **Subtle Integration**: The details given in the premises must not become the central focus of the story. Instead, they should be subtly woven into the main storyline with minimal exposition. **Completeness**: All details and information from the premises must be fully preserved and incorporated into the schema. **No Outcome Leakage**: The schema must not reveal or hint at any consequences or outcomes.

**Context and Continuation Generation**   We then provide the schema and premises to the LLM to generate the final context, following the same three principles outlined above. After producing the context, we combine the consequences and conflicts of Commonsense A and B in various ways and prompt the LLM to generate the corresponding continuations. The continuation that contains both

consequences is designated as the correct option, while any continuation containing a conflict from either commonsense serves as a distractor option.

**Quality Control and Annotation** Beyond using a validator during commonsense generation to guarantee correctness and difficulty, we perform additional quality control on the final test items. To further reduce the probability of random guessing, each distractor option is resampled twice, resulting in a total of ten candidate options per question. We then conduct an automatic sanity check by supplying the full question — together with its associated commonsense pairs — to an LLM. If the model fails to select the correct answer with explicit hints, the item is discarded. For each question, we also use the LLM to annotate local relevance, assigning two labels: **Local Background**: Indicates that the story contains clear elements specific to the culture or environment of the given language. **Local Commonsense**: Indicates that answering the question correctly requires knowledge unique to that language's local culture or context. The prompts used are presented in Appendix B.

## 3.4 HUMAN EVALUATION

We recruited human annotators for each language to evaluate the test questions, with at least three annotators per language. For English, Chinese, Arabic, German, French, Portuguese, and Indonesian, we randomly sampled 100 questions per language. Among them, in 50 questions where the underlying commonsense was provided, human annotators achieved an average accuracy of 92%. In the remaining 50 questions where the commonsense was not given, the average human accuracy dropped to 73%.

We go beyond mere evaluation by performing full human verification and correction of HellaSwag-Ultra to ensure the correctness of all test items. So far, we have completed the verification of 100+ questions each for English, Arabic, German, and French, and released them as a separate dataset called HellaSwagUltra-Gold. Given the large number of languages and questions covered by HellaSwagUltra, we plan to maintain and update this project over the long term. More details of human annotaion and cost are presented in Appendix C.

## 3.5 STATISTICS

Table 2: Statistics of HellaSwagUltra and Verified subset.

| Language | Total | Verified | Local Background | Local Commonsense |
|---|---|---|---|---|
| **ALL** | 62,411 | 766 | 31,237 | 11,984 |
| *Verified* | | | | |
| EN | 958 | 122 | 31 | 8 |
| DE | 1,022 | 231 | 45 | 40 |
| AR | 891 | 168 | 121 | 68 |
| FR | 1,240 | 245 | 32 | 24 |

Table 2 reports statistics for the samples included in HellaSwagUltra. Approximately half of all samples are set against story contexts that exhibit clear language-specific or culturally grounded features, and roughly one-third are annotated as requiring local commonsense knowledge for correct resolution. The table also provides detailed statistics for all languages with human verification. Within the subset of human-verified and annotated samples, English exhibits the lowest proportion of both local backgrounds and local-commonsense requirements. This observation is consistent with the widespread use of English across diverse regions, which makes its content more likely to be perceived as general rather than culturally anchored. German and French similarly show relatively low proportions of items requiring local commonsense, reflecting a closer cultural affinity with English and shared background knowledge. In contrast, Arabic samples display a markedly higher proportion of items that necessitate local commonsense, highlighting the distinct cultural specificity captured in this subset. The agreement between LLM-based annotations and human judgments for both local background and local commonsense exceeds 80%.

Table 3: Model performance on HellaSwagUltra. **ALL** reports the average accuracy across all languages. **HIGH**, **MID**, and **LOW** are averages over high-, mid-, and low-resource languages, respectively. **LB** (Local Background) includes examples whose story context contains target-language-specific cultural elements. **LC** (Local Commonsense) covers examples requiring culture-specific commonsense knowledge, while **GC** (General Commonsense) tests language-agnostic commonsense. **VERIFIED** reports results on HellaSwagUltra-Gold. Base models are evaluated under the Cloze format.

| Model | ALL | HIGH | MID | LOW | LB | LC | GC | VERIFIED |
|---|---|---|---|---|---|---|---|---|
| *Base Models* | | | | | | | | |
| Qwen3-14B-Base | 43.86 | 48.45 | 45.41 | 40.47 | 42.94 | 43.08 | 44.15 | 47.68 |
| Qwen2.5-14B | 41.12 | 48.17 | 42.69 | 36.69 | 40.65 | 40.17 | 41.61 | 47.69 |
| Qwen2.5-32B | 42.25 | 50.53 | 43.58 | 37.52 | 41.58 | 41.52 | 42.55 | 49.34 |
| Qwen2.5-72B | 44.84 | 52.53 | 47.34 | 39.28 | 43.92 | 43.71 | 45.22 | 48.85 |
| gemma-3-12b-pt | 47.77 | 48.09 | 49.92 | 45.67 | 47.08 | 46.61 | 48.27 | 47.51 |
| gemma-3-27b-pt | 50.88 | 51.06 | 53.12 | 48.73 | 50.42 | 50.09 | 51.28 | 49.26 |
| gemma-2-9b | 43.94 | 46.44 | 45.98 | 41.00 | 43.00 | 43.04 | 44.16 | 47.20 |
| gemma-2-27b | 47.15 | 49.25 | 50.24 | 43.41 | 46.13 | 45.76 | 47.63 | 48.56 |
| *Instruct Models* | | | | | | | | |
| Qwen2.5-14B-Instruct | 37.54 | 46.13 | 38.94 | 32.61 | 38.21 | 38.52 | 37.25 | 50.22 |
| Qwen2.5-32B-Instruct | 39.59 | 47.61 | 41.41 | 34.52 | 40.29 | 41.13 | 39.19 | 50.22 |
| Qwen2.5-72B-Instruct | 40.18 | 47.76 | 41.42 | 35.82 | 40.29 | 40.45 | 40.00 | 49.47 |
| gemma-3-12b-it | 34.57 | 36.27 | 35.18 | 33.29 | 35.34 | 34.68 | 34.30 | 40.84 |
| gemma-3-27b-it | 42.02 | 43.55 | 41.66 | 41.70 | 42.19 | 42.28 | 41.94 | 48.32 |
| gemma-2-9b-it | 31.51 | 33.45 | 31.21 | 30.98 | 32.27 | 32.77 | 30.89 | 37.31 |
| gemma-2-27b-it | 37.26 | 40.28 | 37.75 | 35.54 | 37.50 | 37.64 | 36.98 | 45.01 |
| *Proprietary Model* | | | | | | | | |
| GPT-4o | 40.34 | 47.17 | 41.87 | 36.02 | 40.29 | 40.11 | 40.40 | 50.64 |
| Claude Sonnet 4 | 63.12 | 65.58 | 64.46 | 60.86 | 62.78 | 62.11 | 63.52 | 60.14 |
| Claude Opus 4 | 64.53 | 66.28 | 65.31 | 63.07 | 63.75 | 63.06 | 65.11 | 59.57 |
| Gemini-2.5-Pro | 72.13 | 70.31 | 73.11 | 72.00 | 71.71 | 71.30 | 72.60 | 62.53 |

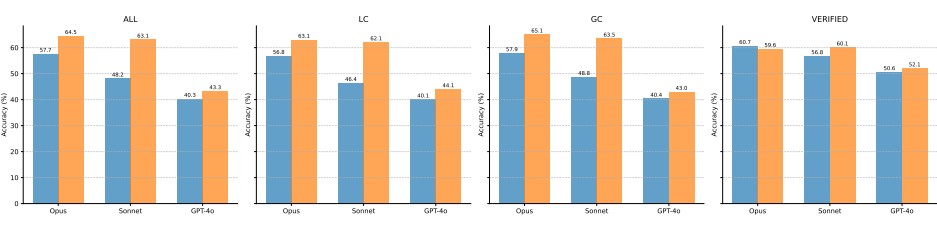

Figure 3: Model Performance with and without thinking.

## 4 EVALUATION

### 4.1 SETUP

**Task Format** HellaSwagUltra retains the core task format of HellaSwag: selecting the most plausible continuation given a context. This design preserves the natural fluency of the text and aligns closely with the causal LLM training objective, ensuring stable evaluation. For base models, we adopt a Cloze-style setup (Clark et al., 2018), computing the perplexity (PPL) for each candidate continuation and selecting the one with the lowest PPL as the model's choice. For instruction-tuned models, we aim to measure their answering ability directly. We present the question as a standard multiple-choice problem with a simple instruction — "Which option is the most plausible continuation?" — provided in two variants: English and a localized version in the target language. In this paper, we use the localized instruction to more accurately reflect performance in multilingual settings.

**Metric** We use simple evaluation metrics to ensure that HellaSwagUltra can be easily integrated into any LLM evaluation framework. For base models tested in the Cloze format, we report accuracy based on the model's selection of the option with the lowest perplexity. For instruction-tuned mod-

els, we report Exact Match (EM), indicating whether the model's generated answer exactly matches the correct option.

**Models**   We evaluate open-source base models known for their strong multilingual performance, including the Qwen (Qwen et al., 2025; Yang et al., 2025) and Gemma families (Team et al., 2024b; 2025). Likewise, the instruction-tuned variants of these model families are also included in our evaluation. It is important to note, however, that the evaluation protocols for base models and instruction-tuned models differ, as described above. In addition to open-source models, we also benchmark several of the most capable closed-source models currently available, including GPT-4o (OpenAI et al., 2024), Claude Opus 4 and Claude Sonnet 4.

### 4.2   RESULTS

Table 3 presents the evaluation results. We observe that all evaluated models perform suboptimally on HellaSwagUltra, indicating that our dataset presents a sufficient level of challenge and effectively mitigates the saturation observed in many existing commonsense benchmarks. Importantly, this increased difficulty arises from more demanding commonsense reasoning requirements rather than from the use of overly complex instructions or evaluation metrics. Within each model family, we observe a clear scaling trend: larger models consistently achieve better performance. Across languages, all models exhibit a noticeable performance drop on low-resource

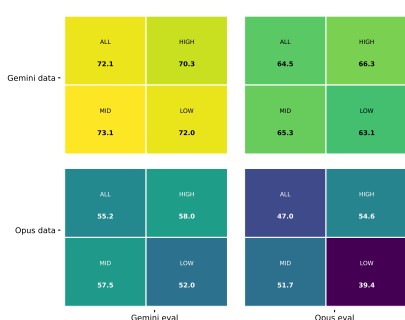

Figure 4: Cross-evaluation of datasets generated by different models.

languages. Among the open-source families, the Gemma series demonstrates relatively balanced performance across languages, whereas the Qwen series shows a more pronounced gap between high- and low-resource languages. As expected, closed-source models generally outperform their open-source counterparts. Notably, GPT-4o, which is not a reasoning model, shows a substantial performance gap compared to Claude models.

### 4.3   EFFECT OF THINKING

We evaluate the impact of enabling thinking mode on model performance on HellaSwag. For Opus and Sonnet, we compare their performance with thinking mode enabled and disabled. For GPT-4o, which is not a reasoning model by default, we apply a Chain-of-Thought (CoT) prompt to encourage reasoning before generating an answer and compare this to its direct output. Figure 3 illustrates the result. All three models show substantial performance gains on the full dataset when thinking mode is enabled, while the improvement is smaller on the Verified subset, with Opus showing a slight decline. On the local-commonsense subset, Sonnet and GPT-4o exhibit larger gains compared to their performance on the non-local-commonsense subset.

### 4.4   POTENTIAL MODEL BIAS STUDY

To investigate whether the LLM used for data generation would gain an advantage when evaluated on that data, we additionally used Claude Opus 4 to generate 300 samples per language with the same pipeline, and evaluated both Gemini-2.5-Pro and Claude Opus 4 on this data. Figure 4 depicts the cross-evaluation results. It can be observed that Opus does not exhibit an advantage on the data generated by itself. Therefore, the bias introduced by the model is not significant, and any potential risk will be further mitigated through human correction. Gemini shows a clear performance advantage over Opus, so we chose Gemini-2.5-Pro to generate HellaSwagUltra.

## 5   CONCLUSION

Faced with the saturation of commonsense reasoning benchmarks and the scarcity of multilingual resources, this paper introduces HellaSwagUltra, a new dataset supporting over 60 languages, focused on challenging commonsense reasoning and language local knowledge.

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

## A  Language Coverage

Table 4 presents the language covered by HellaSwagUltra. Considering only native speakers, these languages cover over 60% of the global population. When including second-language speakers, the coverage exceeds 99% worldwide.

Table 4: Languages sorted by native speakers and ratios in Common Crawl (HIGH at left, MID center, LOW right)

| Code | Name | Speakers | Tokens | Code | Name | Speakers | Tokens | Code | Name | Speakers | Tokens |
|------|------|----------|--------|------|------|----------|--------|------|------|----------|--------|
| zh | Chinese | 1390M | 6.34% | vi | Vietnamese | 86M | 1.35% | hi | Hindi | 345M | 0.31% |
| es | Spanish | 484M | 4.14% | tr | Turkish | 85M | 0.98% | bn | Bengali | 242M | 0.18% |
| ar | Arabic | 411M | 0.78% | ms | Malay | 82M | 0.03% | mr | Marathi | 83M | 0.04% |
| en | English | 390M | 42.62% | ur | Urdu | 78M | 0.04% | te | Telugu | 83M | 0.03% |
| pt | Portuguese | 250M | 1.51% | id | Indonesian | 75M | 1.05% | ta | Tamil | 79M | 0.09% |
| ru | Russian | 145M | 9.16% | fa | Persian | 65M | 0.79% | jv | Javanese | 69M | 0.00% |
| ja | Japanese | 124M | 4.72% | pl | Polish | 38M | 1.69% | gu | Gujarati | 58M | 0.03% |
| ko | Korean | 81M | 0.84% | th | Thai | 38M | 0.64% | my | Burmese | 33M | 0.03% |
| de | German | 76M | 5.21% | uk | Ukrainian | 32M | 0.60% | pa | Punjabi | 32M | 0.01% |
| fr | French | 74M | 4.10% | ro | Romanian | 24M | 0.64% | tl | Tagalog | 28M | 0.02% |
| it | Italian | 63M | 2.33% | nl | Dutch | 23M | 1.57% | uz | Uzbek | 27M | 0.01% |
|  |  |  |  | el | Greek | 12M | 0.69% | az | Azerbaijani | 24M | 0.10% |
|  |  |  |  | bg | Bulgarian | 8M | 0.32% | ceb | Cebuano | 21M | 0.00% |
|  |  |  |  | hr | Croatian | 5.1M | 0.24% | sw | Swahili | 16M | 0.01% |
|  |  |  |  | sk | Slovak | 5M | 0.35% | km | Khmer | 16M | 0.02% |
|  |  |  |  | he | Hebrew | 5M | 0.27% | sq | Albanian | 7.5M | 0.05% |
|  |  |  |  | lt | Lithuanian | 2.8M | 0.18% | af | Afrikaans | 7M | 0.01% |
|  |  |  |  | lv | Latvian | 1.75M | 0.10% | no | Norwegian | 5.3M | 0.37% |
|  |  |  |  | et | Estonian | 1.1M | 0.14% | da | Danish | 5M | 0.36% |
|  |  |  |  |  |  |  |  | fi | Finnish | 5M | 0.41% |
|  |  |  |  |  |  |  |  | is | Icelandic | 0.314M | 0.04% |
|  |  |  |  |  |  |  |  | ga | Irish | — | 0.01% |

## B  Prompts

The prompts used in data collection are as follows.

## Commonsense Generation

We are designing a local commonsense knowledge in {LANGUAGE} that will be subtly implied in a story. The `conflict_detail` is a subtle narrative detail that may hint that the commonsense has been violated. Later, this will be used to generate an incorrect story continuation.

**[Reference WiKi]**

{WIKITEXT}

**[TASK]**

Invent a new piece of local commonsense knowledge in {LANGUAGE}.

Return a single JSON object using this exact schema with the content in {LANGUAGE}:

**concept** : short noun phrase (1–3 words)

**description** : one sentence explaining the concept

**premise** : a brief and typical sign or condition under which the consequence usually holds (no need for full coverage)

- The `premise` is a specific condition or context.
- The `premise` should **not include or imply the consequence itself** — it must be a distinct and self-contained condition, not a restatement or soft prediction of the result.
- The `premise` must be both necessary **and** sufficient for the consequence.

**consequence** : what is expected under the premise

**conflict_detail** : a subtle, objective event or scene implying the consequence may have been violated (do NOT state the violation or directly revise the property)

- `conflict_detail` should hint at a breach indirectly and be subtle.
- Describe only observable facts, actions, or physical details.
- Do **NOT** mention the items or properties that appear in the consequence.
- Do **NOT** overemphasize or elaborate on the conflicting detail.
- Do **NOT** emphasize or mention what did *not* happened.
- Do **NOT** justify, rationalize, or explain the detail that contradicts commonsense.
- Do **NOT** use negation words (e.g., "not", "no", "never", "didn't", "hasn't", "without", "failed to").
- Do **NOT** use speculative or ambiguous expressions (e.g., "seems", "appears", "perhaps", "maybe", "apparently", "as if").
- Do **NOT** use contrastive words like "but", "however".
- Do **NOT** include thoughts, feelings.

**Quality requirements:**

The commonsense must be strong and widely accepted: in ordinary contexts it should hold with certainty; violating it should make the scenario feel blatantly unrealistic or jarringly wrong to a knowledgeable reader.

Given the `premise`, the `conflict_details` should be virtually impossible to happen in reality.

Ensure **diversity** from the reference examples, the topic **MUST BE DIFFERENT** from any of the reference examples.

Consider diverse types and maintain a numerical balance between **traditional and modern**, **cultural and scientific**, **local and worldwide** commonsenses.

Both the `premise` and the `consequence` should have a **moderate level of abstraction**: avoid overly specific names, locations, or one-time events. The statements should apply to many plausible real-world situations.

**Reference examples:**

{examples}

```
Commonsense Revision
```

We are designing a local commonsense knowledge in {LANGUAGE} that will be subtly implied in a story. The `conflict_detail` is a subtle narrative detail that may hint that the commonsense has been violated. Later, this will be used to generate an incorrect story continuation.

**[REFERENCE EXAMPLES]**
{examples}
**[PREVIOUS VERSION]**
{previous_json_pretty}
**[VALIDATOR COMMENTS]**
{feedback_rules_block}
**[TASK]**
Please revise the above commonsense item to better follow the rules and address the validator feedback.

Return a single JSON object using this exact schema with the content in {LANGUAGE}:

**concept** : short noun phrase (1–3 words)

**description** : one sentence explaining the concept

**premise** : a brief and typical sign or condition under which the consequence usually holds (no need for full coverage)

- The `premise` is a specific condition or context.

- The `premise` should **not include or imply the consequence itself** — it must be a distinct and self-contained condition, not a restatement or soft prediction of the result.

- The `premise` must be both necessary **and** sufficient for the consequence.

**consequence** : what is expected under the premise

**conflict_detail** : a subtle, objective event or scene implying the consequence may have been violated (do NOT state the violation or directly revise the property)

- `conflict_detail` should hint at a breach indirectly and be subtle.

- Describe only observable facts, actions, or physical details.

- Do **NOT** mention the items or properties that appear in the consequence.

- Do **NOT** overemphasize or elaborate on the conflicting detail.

- Do **NOT** emphasize or mention what did not happend.

- Do **NOT** justify, rationalize, or explain the detail that contradicts commonsense.

- Do **NOT** use negation words (e.g., 'not', 'no', 'never', 'didn't', 'hasn't', 'without', 'failed to').

- Do **NOT** use speculative or ambiguous expressions (e.g., 'seems', 'appears', 'perhaps', 'maybe', 'apparently', 'as if').

- Do **NOT** use contrastive words like 'but', 'however'.

- Do **NOT** include thoughts, feelings.

**Quality requirements:**

The commonsense must be strong and widely accepted: in ordinary contexts it should hold with certainty; violating it should make the scenario feel blatantly unrealistic or jarringly wrong to a knowledgeable reader.

Given the `premise`, the `conflict_details` should be virtually impossible to happen in reality.

Ensure **diversity** from the reference examples, the topic **MUST BE DIFFERENT** from any of the reference examples.

Consider diverse types and maintain a numerical balance between **traditional and modern**, **cultural and scientific**, **local and worldwide** commonsenses.

Both the `premise` and the `consequence` should have a **moderate level of abstraction**: avoid overly specific names, locations, or one-time events. The statements should apply to many plausible real-world situations.

---

**Commonsense Validator**

We are designing a **{LANGUAGE} commonsense knowledge** item that will be subtly implied in a story. `'description'` explains the piece of commonsense knowledge. `'premise'` and `'consequence'` denote, respectively, the preconditions under which this commonsense holds and its expected result. `'conflict_detail'` is a subtle detail that hints at a violation of that result. Your job is to evaluate whether the defined commonsense knowledge and the `'conflict_detail'` are valid and well-formed for this purpose based on the following rule.
**[RULE]**
{RULE_TEXT}
**[GUIDELINE]**
{GUIDELINE_TEXT}
**[COMMONSENSE UNDER REVIEW]**
{COMMONSENSE}
**Respond in JSON with:**

```
{
   "comment": string,
   "decision": "pass" | "fail"
}
```

---

```
Story Schema Builder
```

We are designing a **beginning** of a short realistic story in {LANGUAGE} that integrates both a visible storyline and a hidden layer.
You are given some subtle details.
Your task is to design the schema.
**Important constraints:**

- The subtle details must **NOT** be the focus and mainstream of the story, nor the characters' main activity.

- Subtly weave the subtle details into the main storyline with minimal exposition.

- Ensure the information in the provided details is **accurately** and **completely** included.

- Do not reveal or speculate about the continuation of the subtle details.

**Please provide the following fields:**

1. **Scene**: Where does the story take place? Describe the physical and social setting briefly.

2. **Characters**: List 2–3 people involved in the scene, with their names, roles, motivations, features, characteristics, relationship, etc.

3. **Objects**: Any key items or tools present in the scene.

4. **Situation description**: A short paragraph (3–4 sentences) describing what's happening, *without stating* the commonsense.

5. **Goal or activity**: What is the apparent goal of the characters in the scene?

6. **Visible tension or obstacle**: Is there any small conflict or uncertainty that drives the scene forward?

You can add more fields.
Return all fields in plain text in {LANGUAGE}.
**Example**

```
Subtle details: Two people are sitting in a café and talking.
Output in English:
```
**Scene**: A quiet café in the afternoon.
**Characters**: Lisa, a journalist; Mark, her childhood friend.
**Objects**: Coffee cups, a notepad, a scarf hanging behind Lisa's chair.
**Situation description**: Lisa leans across the table, her voice low as she asks Mark about the article. He listens, occasionally glancing at the entrance. The café hums softly around them.
**Goal or activity**: They are catching up and discussing a sensitive interview.
**Visible tension or obstacle**: Lisa is worried someone may overhear them.

**Now you generate:**
```
Subtle details:
```
{premise}

```
Output in {LANGUAGE}:
```

**Story Context Generator**

Given a designed schema, write a beginning of a short, realistic story in {LANGUAGE}.
**Constraints:**

- The story **must include natural and appropriate mentions** of all people, objects, or situations referenced in the provided schema, but they should appear **organically and with narrative motivation** — not feel forced or inserted just to match the fact.

- The tone should be grounded, realistic, and coherent.

- The story should reflect the described **scene, characters, objects, activity, and visible tension** through concrete actions, sensory details, or dialogue.

- No explaining or summarizing; let the details emerge naturally.

- Subtly weave the provided subtle details into the main storyline with minimal exposition. Ensure all the key information is **accurately** and **completely** included.

- The embedded subtle details must **NOT** be the focus and mainstream of the story, nor the characters' main activity.

- Focus on objective, observable descriptions of actions, settings, and dialogue.

- Do **NOT** add thoughts, feelings, or commentary.

- Do **NOT** mention what did **not** happen.

- Do **NOT** use speculative or ambiguous expressions (e.g., "seems", "appears", "perhaps", "maybe", "apparently", "as if").

- The story should be in 4–5 sentences.

**Example**

**Schema:**
**Scene**: A quiet café in the afternoon.
**Characters**: Lisa, a journalist; Mark, her childhood friend.
**Objects**: Coffee cups, a notepad, a scarf hanging behind Lisa's chair.
**Situation description**: Lisa leans across the table, her voice low as she asks Mark about the article. He listens, occasionally glancing at the entrance. The café hums softly around them.
**Goal or activity**: They are catching up and discussing a sensitive interview.
**Visible tension or obstacle**: Lisa is worried someone may overhear them.

**Subtle details:**
Two people are sitting in a café and talking.

**Story in English:**
Lisa lowered her voice, scribbling something in her notepad as Mark leaned in. The scarf behind her chair fluttered slightly as the door opened. He looked up, eyes scanning the new arrival. "Do you think they're listening?" she whispered.

**Now complete the story based on the following schema:**
{schema}
**Subtle details:**
{details}
**Story in {LANGUAGE}:**

---

### Story Continuation A (Positive)

You are given the schema of a short story in {LANGUAGE}, a beginning and some subtle details.

**Your task is to write a plausible continuation of the story:**

- The continuation must naturally **follow from the story so far**, not repeat or revise the story.
- Subtly weave the follow-up of the provided details into the main storyline with minimal exposition.
- Do not justify, rationalize, or explain the details.
- Focus on objective, observable descriptions of actions, settings, and dialogue.
- Do **NOT** add thoughts, feelings, or commentary.
- Do **NOT** mention events or outcomes that did **not** happen — focus on what is occurring in the scene.
- Do **NOT** use negation words (e.g., "not", "no", "never", "didn't", "hasn't", "without", "failed to").
- Do **NOT** use speculative or ambiguous expressions (e.g., "seems", "appears", "perhaps", "maybe", "apparently", "as if").
- The continuation should be in **1–2 sentences**.

**Example**

**Schema:**
**Scene**: A quiet café in the afternoon.
**Characters**: Lisa, a journalist; Mark, her childhood friend.
**Objects**: Coffee cups, a notepad, a scarf hanging behind Lisa's chair.
**Situation description**: Lisa leans across the table, her voice low as she asks Mark about the article. He listens, occasionally glancing at the entrance. The café hums softly around them.
**Goal or activity**: They are catching up and discussing a sensitive interview.
**Visible tension or obstacle**: Lisa is worried someone may overhear them.

**Story so far:**
Lisa lowered her voice, scribbling something in her notepad as Mark leaned in. The scarf behind her chair fluttered slightly as the door opened. He looked up, eyes scanning the new arrival. "Do you think they're listening?" she whispered.

**Subtle details:**
One cannot see what is happening behind himself/herself.

**Continuation in English:**
Mark shook his head subtly, his eyes drifting past Lisa toward the window. A delivery man stepped inside, pausing to check the receipt in his hand.

**Now you generate**
**Schema:**
{schema}
**Story so far:**
{story}
**Subtle details:**
{details}
**Continuation in {LANGUAGE}:**

---

---

**Story Continuation B (Negative)**

You are given the schema of a short story in {LANGUAGE}, a beginning, a continuation A and some subtle details.

**Your task is to write a nuanced different continuation B of the story:**

- The continuation must naturally **follow from the story so far**, not repeat or revise the story.

- Subtly but **accurately** weave all of the "follow-up" and "conflict" details provided into the main storyline with **minimal exposition**.

- Do **not** include, mention, explain, or describe the knowledge and premise in the provided details.

- Do **not** justify, rationalize, or explain the follow-up and conflict in the provided details.

- Focus on objective, observable descriptions of actions, settings, and dialogue.

- Do **NOT** add thoughts, feelings, or commentary.

- Do **NOT** mention events or outcomes that did **not** happen — focus on what is occurring in the scene.

- Do **NOT** use negation words (e.g., "not", "no", "never", "didn't", "hasn't", "without", "failed to").

- Do **NOT** use speculative or ambiguous expressions (e.g., "seems", "appears", "perhaps", "maybe", "apparently", "as if").

- The continuation should be in **2–3 sentences**.

**Example**

**Schema:**
**Scene**: A quiet café in the afternoon.
**Characters**: Lisa, a journalist; Mark, her childhood friend.
**Objects**: Coffee cups, a notepad, a scarf hanging behind Lisa's chair.
**Situation description**: Lisa leans across the table, her voice low as she asks Mark about the article. He listens, occasionally glancing at the entrance. The café hums softly around them.
**Goal or activity**: They are catching up and discussing a sensitive interview.
**Visible tension or obstacle**: Lisa is worried someone may overhear them.

**Story so far:**
Lisa lowered her voice, scribbling something in her notepad as Mark leaned in. The scarf behind her chair fluttered slightly as the door opened. He looked up, eyes scanning the new arrival. "Do you think they're listening?" she whispered.

**Continuation A:**
Mark shook his head subtly, his eyes drifting past Lisa toward the window. A delivery man stepped inside, pausing to check the receipt in his hand.

**Subtle details:**
As they chatted, one of them quietly described the suspicious figure sneaking up behind himself.

**Continuation B in English:**
Mark nodded toward the hallway. "Someone just slipped behind me," Lisa said, frowning.

**Now you generate**
**Schema:**
{schema}
**Story so far:**
{story}
**Continuation A:**
{silver}
**Subtle details:**
{details}
**Continuation B in {LANGUAGE}:**

---

## C    HUMAN EVALUATION AND COST

We recruited human annotators who were required to hold at least a college degree, demonstrate C1-level English proficiency (or an equivalent certification), and be native speakers of the languages they were assigned to evaluate. Annotators were paid at an hourly rate of $16, with a maximum of 8 working hours per day. To date, the total cost of human annotation is approximately $19,200.

In addition, the API cost for LLM calls during data collection is approximately $36,800.

