# OpenReview forum: "Think you have Solved Commonsense Reasoning? Try HellaswagUltra"
_ICLR.cc/2026/Conference — ICLR 2026 Conference Withdrawn Submission_

### Official Review · Reviewer_K2vh · 2025-10-25

**Soundness:** 2
**Presentation:** 3
**Contribution:** 3
**Rating:** 4
**Confidence:** 3

**Summary:**

This paper introduces HellaSwagUltra, a large-scale multilingual benchmark for commonsense reasoning that spans over 60 languages and 62,000 instances. It embeds multiple implicit, culture-specific, and general commonsense facts within narrative contexts featuring challenging distractor options. The dataset is built through an automated LLM-driven pipeline that extracts and validates culturally grounded knowledge from Wikipedia, transforms it into contextualized stories, and filters candidate answers. A representative subset, HellaSwagUltra-Gold, is extensively annotated and human-verified.

**Strengths:**

1. The paper convincingly identifies the saturation of existing commonsense benchmarks such as HellaSwag and StoryCloze, and rightly points out that translation-based multilingual benchmarks risk introducing cultural bias. The proposed HellaSwagUltra addresses both issues by embedding culturally grounded and language-specific commonsense knowledge, thereby tackling the dual challenges of benchmark saturation and limited linguistic coverage.

2. The scale and linguistic breadth of the dataset are genuinely impressive. With more than sixty languages represented, this resource could become a cornerstone for future research on multilingual commonsense reasoning and cross-cultural evaluation.

3. The construction pipeline is clearly structured and easy to follow. Each stage is presented in a transparent manner that suggests good reproducibility.

4. The evaluation section is also thorough. It considers multiple model families, including state-of-the-art LLMs, and thoughtfully examines thinking or reasoning modes, which adds an extra layer of depth to the analysis.

**Weaknesses:**

1. While the data construction pipeline is well described, the paper lacks quantitative analysis showing how much each stage contributes to the final benchmark quality. An ablation study or step-wise evaluation would greatly strengthen the claims of robustness and reliability.

2. Similarly, although the overall results are clearly reported, there is little insight into where and why models fail. It would be valuable to understand which types of distractors are most confusing or which specific languages and domains pose the greatest difficulty.

3. The paper would also benefit from a more detailed qualitative error analysis—presenting concrete examples of model mistakes to clarify what kinds of commonsense reasoning remain challenging.

4. Information about the human annotation process could be expanded. In particular, inter-annotator agreement, sample sizes per language, and detailed evaluation protocols are only briefly mentioned and should be made more explicit to increase confidence in the dataset’s consistency.

5. The discussion would be richer if it included qualitative examples or case studies illustrating how certain linguistic or cultural subtleties affect model behavior.

**Questions:**

I understand that conducting detailed per-language analysis can be difficult given the dataset’s size. Still, it would be helpful to know:

1. Did the authors observe any language-specific artifacts or systematic biases during data generation or evaluation?

2. Are there measurable differences in difficulty or accuracy across languages, and if so, what factors (e.g., data resources, linguistic structure) might explain them?

---

### Official Review · Reviewer_7Gex · 2025-10-28

**Soundness:** 2
**Presentation:** 2
**Contribution:** 2
**Rating:** 2
**Confidence:** 4

**Summary:**

This paper introduces HellaSwagUltra, a large-scale benchmark for multilingual commonsense reasoning. It is designed to address the saturation of existing English-centric benchmarks and the lack of culturally-aware evaluations in other languages. The benchmark covers over 60 languages and is built using an automated data construction pipeline. Experiments demonstrate the benchmark's difficulty: even the strongest proprietary models like Gemini-2.5-Pro achieve only 62.53% accuracy on the human-verified subset (HellaSwagUltra-Gold) , while other models perform near 40-50%.

**Strengths:**

1.The paper is easy to read, featuring informative figures that clearly illustrate the data construction pipeline and provide concrete examples comparing HellaSwagUltra to other benchmarks .
2. It places an emphasis on cultural differences, grounding the benchmark in each language's native culture and social context and ensuring it includes local cultural knowledge.

**Weaknesses:**

1．This paper vasts majority of data lacks human verification, casting doubt on reliability. The authors claim the benchmark contains 62,411 total samples , but the human-verified "HellaSwagUltra-Gold" subset consists of only 766 samples.These 766 verified samples cover only 4 of the 60+ languages (English, German, Arabic, and French). This means that over 98.8% of the dataset (61,645 samples) has not undergone any human review. However, the paper's main experimental results, such as the cross-language average in the "ALL" column of Table 3, are based on this overwhelmingly unverified full dataset. This calls the reliability of the results into question. We cannot be certain whether models perform poorly on these unverified samples because the task is genuinely challenging, or because the data contains LLM-generated noise, factual errors, nonsensical conflict options, or other artifacts.
2. The benchmark is over-reliance on a single LLM for the entire construction pipeline. The authors state that the entire data construction process, from knowledge extraction and structured commonsense generation to story design and final context/option generation, relied on Gemini-2.5-Pro throughout.This approach introduces a significant risk of systemic bias. The benchmark may not be testing for general, cross-cultural commonsense, but rather testing the extent to which other models can replicate the specific knowledge base, generation style, and potential biases of Gemini-2.5-Pro.
3. Lack of human verification amplifies concerns about the "LLM Validator". Prior to the (extremely limited) human annotation, the pipeline relies primarily on an "LLM-based validator" and an "automatic sanity check" to ensure quality.This is essentially asking an LLM (Gemini-2.5-Pro) to review its own work. Given the first weakness (that 98.8% of the data is unverified), it is highly questionable whether this "self-check" loop is sufficient to guarantee the correctness of the commonsense, the validity of the conflicts, or the singular plausibility of the correct answer.
4. The paper is lack of analysis for the performance drop in instruct models. The data in Table 3 reveals a consistent and interesting trend: Base Models generally outperform their corresponding Instruct-Tuned Models (e.g., in the Gemma families). However, the paper merely presents this data without any discussion or analysis. This is an omission, as it misses the opportunity to discuss whether current instruction tuning (SFT/RLHF) might be harming the models' underlying causal language modeling capabilities or their judgment of plausibility in natural language.

**Questions:**

1.The main results in Table 3 (the "ALL" column) are based on the full 62,411-sample dataset, yet the human-verified "HellaSwagUltra-Gold" subset contains only 766 samples. This means over 98.8% of the main benchmark is unverified. How can we be confident that the low model scores on the full dataset are due to the task's genuine difficulty, and not due to LLM-generated noise, factual errors, or flawed distractors in this massive unverified portion?
2.The paper presents "Local Commonsense" (LC) as a key contribution. However, the provided Chinese high-speed rail example (e.g., trains are punctual, tracks are continuously welded ) appears to be factual local knowledge rather than commonsense reasoning. How does the benchmark distinguish between testing culturally-specific facts (which could be memorized) and testing culturally-specific reasoning patterns?
3.The "Potential Model Bias Study" (Section 4.4) uses only 300 samples per language generated by Claude Opus 4 to conclude that generator bias is "not significant". Is this limited-scale, two-model comparison truly sufficient to rule out the possibility that the benchmark's difficulty and structure are heavily biased by the specific design choices and prompting methods of the Gemini-2.5-Pro-based pipeline?
4.The caption for Table 3 should be placed below the table.
5.The font size in the figures is too small and difficult to read.
6.What do the different colored sections in Figure 1 represent?

---

### Official Review · Reviewer_N9zv · 2025-10-31

**Soundness:** 3
**Presentation:** 3
**Contribution:** 3
**Rating:** 6
**Confidence:** 4

**Summary:**

This paper introduces HellaSwagUltra, a new benchmark for multilingual commonsense reasoning. The core contribution is a large-scale dataset spanning over 60 languages, specifically designed to test reasoning about local culture and commonsense knowledge. The authors present an automated pipeline for generating culturally-grounded story scenarios with implicit commonsense questions, which proves difficult even for state-of-the-art models. A human-verified subset, HellaSwagUltra-Gold, is also provided for high-fidelity evaluation.

**Strengths:**

The premise of creating a challenging, non-saturated multilingual commonsense benchmark that focuses on local culture is novel and valuable for the field. A significant contribution is the automated data construction pipeline, which is well-designed, scalable, and makes the benchmark a sustainable and continuously expandable resource. The inclusion of a human-annotated "gold" standard subset adds a layer of reliability.

**Weaknesses:**

the work could be improved by a more thorough experimental analysis. The current evaluations feel somewhat shallow for a benchmark of this scope. For example, the paper does not deeply investigate why instruction-tuned models perform significantly worse than base models, which might suggest that the mixed evaluation formats (cloze vs. multiple-choice) are not directly comparable. More critically, the central claim of testing local commonsense is not conclusively supported by the results. The performance difference between the General Commonsense (GC) and Local Commonsense (LC) splits is marginal for top models, which questions whether the dataset effectively isolates culturally specific reasoning challenges. The evaluation does not manage to disentangle failures of linguistic understanding from deficits in cultural knowledge.

**Questions:**

Have the authors considered an analysis to disentangle the language understanding aspect from the cultural knowledge aspect?

---

### Official Review · Reviewer_NeoY · 2025-11-01

**Soundness:** 1
**Presentation:** 1
**Contribution:** 1
**Rating:** 2
**Confidence:** 5

**Summary:**

This paper introduces HellaSwagUltra, a multilingual commonsense reasoning benchmark covering 60+ languages with 62,411 test instances. The authors aim to address three key challenges in existing benchmarks: (1) saturation of English-only benchmarks like HellaSwag where strong models achieve near-perfect scores, (2) limited multilingual coverage with insufficient attention to local cultural knowledge, and (3) difficulty in designing challenging yet fair evaluation tasks that work across both small and large models. During benchmark construction, the paper proposes a fully automated three-stage data construction pipeline, including knowledge extraction stage, structured commonsense generation stage, and test rollout stage. Beyond the automated pipeline, the authors conduct human evaluation and verification. For experiments, the paper evaluates various open-source base models (Qwen and Gemma families ranging from 9B to 72B parameters) using a Cloze-style setup where perplexity is computed for each candidate continuation, as well as instruction-tuned variants of these models and closed-source models (GPT-4o, Claude Opus 4, Claude Sonnet 4, Gemini-2.5-Pro) using a standard multiple-choice format with localized instructions.

**Strengths:**

1. The paper tackles a significant gap in multilingual NLP evaluation by focusing on commonsense reasoning across 60+ languages with explicit emphasis on local cultural knowledge, moving beyond the saturation of English-only benchmarks and the limitations of simple translation-based multilingual datasets.

**Weaknesses:**

1. As a paper whose primary contribution is the proposed dataset, it fails to provide any concrete examples of dataset construction or model responses, only a few prompts used for data generation are included. The anonymous link provided contains merely a brief introduction. I do not even know which 61 languages are included in Table 1, and Table 4 only reports results for 52 languages. For a dataset-centric paper, I strongly recommend including several representative examples and corresponding model outputs. In its current form, the work raises two major concerns: doubts about the dataset’s actual existence and skepticism regarding its data quality.

2. The data generation, validation, and evaluation processes all rely on Gemini-2.5-Pro, which inherently introduces a bias toward producing samples that Gemini-2.5-Pro can answer correctly. Consistent with this, Gemini-2.5-Pro achieves the best results in the experiments, likely due to this circular dependence.

3. The paper's main claimed contribution is the notion of "local commonsense", yet the definition is vague. What qualifies as local? For example, is "not putting metal in a microwave" considered universal commonsense or something that requires local education?

4. Table 2 presents evaluation results for only four language subsets, implying that the dataset is still an incomplete prototype. The paper claims to have automatically generated a large-scale dataset of 62k samples using LLMs, but only 766 verified samples are actually reliable, making this claim potentially misleading. Moreover, the results in Table 3 show inconsistent trends between "ALL" and "VERIFIED" subsets (Gemini drops by 9.6%, whereas GPT-4o improves by 10.3%), yet the authors do not discuss these anomalies.

**Questions:**

Please address the above weaknesses.

---

### Note · Authors · 2025-12-05

I have read and agree with the venue's withdrawal policy on behalf of myself and my co-authors.